# Novel Synthetic Routes to Prepare Biologically Active Quinoxalines and Their Derivatives: A Synthetic Review for the Last Two Decades

**DOI:** 10.3390/molecules26041055

**Published:** 2021-02-18

**Authors:** Hena Khatoon, Emilia Abdulmalek

**Affiliations:** 1Department of Chemistry, Faculty of Science, Universiti Putra Malaysia, 43400 UPM Serdang, Selangor Darul Ehsan, Malaysia; 2Integrated Chemical BioPhysics Research, Faculty of Science, University Putra Malaysia, 43400 UPM Serdang, Selangor Darul Ehsan, Malaysia

**Keywords:** quinoxaline, SAR, biological applications, green chemistry

## Abstract

Quinoxalines, a class of *N*-heterocyclic compounds, are important biological agents, and a significant amount of research activity has been directed towards this class. They have several prominent pharmacological effects like antifungal, antibacterial, antiviral, and antimicrobial. Quinoxaline derivatives have diverse therapeutic uses and have become the crucial component in drugs used to treat cancerous cells, AIDS, plant viruses, schizophrenia, certifying them a great future in medicinal chemistry. Due to the current pandemic situation caused by SARS-COVID 19, it has become essential to synthesize drugs to combat deadly pathogens (bacteria, fungi, viruses) for now and near future. Since quinoxalines is an essential moiety to treat infectious diseases, numerous synthetic routes have been developed by researchers, with a prime focus on green chemistry and cost-effective methods. This review paper highlights the various synthetic routes to prepare quinoxaline and its derivatives, covering the literature for the last two decades. A total of 31 schemes have been explained using the green chemistry approach, cost-effective methods, and quinoxaline derivatives’ therapeutic uses.

## 1. Introduction

Quinoxaline is defined as a weakly basic bi-cyclic compound C_8_H_6_N_2_, having fused benzene and pyrazine rings. Quinoxaline is a nitrogen-containing heterocyclic compound and is an indispensable structural unit for both chemists and biochemists. The structure is as shown in Figure 1.

Pthalazine, Quinazolines, and Cinnolenes are similar to Quinoxaline [1], as shown in Figure 2.

Quinoxaline is a low melting solid and is miscible in water. It is a weak base and can form salts with acids. The synthesis of quinoxaline has been extensively studied for the last two decades. A very primitive and effective method to derive quinoxaline is the condensation reaction between ortho phenylenediamine and dicarbonyl compounds [2,3]. This procedure requires a high temperature, a strong acid catalyst, and long hours of heating. Recently, there has been a tremendous increase in green methodology to synthesize quinoxalines such as recyclable catalyst [4], one-pot synthesis [5], microwave-assisted synthesis [6,7], and reactions in an aqueous medium [8].

A little modification in their structure brings different moieties, which has the remarkable pharmacological effect of fighting different diseases with little side effects. Several quinoxaline derivatives have been tested in the last two decades and produced anti-inflammatory [9], antimalarial [10], antidepressants [11], antiviral [12], antimicrobial activity [13] as antifungal and antibacterial agents. The antibacterial activity [13] comprises both gram-positive and gram-negative bacteria, including mycobacterium species. Some derivatives of quinoxaline-1,4-di-*N*-oxide have been shown to inhibit M. tuberculosis to a rate of 99 to 100% [14]. The antifungal properties of quinoxalines have been tested against numerous fungal species, and researchers have reported 2-sulphonyl quinoxalines, 3-[(alkylthio) methyl] quinoxaline-1-oxide derivatives, and also pyrazolo quinoxalines as compounds with high antifungal activity [15].

Quinoxaline is a vital component in anticancer drugs, with promising results [15]. The pursuit for anticancer drugs led to a breakthrough in the anticancer activity of several quinoxaline-1,4-di-*N*-oxide derivatives. A new series of 2-alkylcarbonyl and 2-benzoyl-3-trifluoromethyl quinoxaline-1,4-di-*N*-oxide have been reported to show in vitro tumor activity against three cell line panels comprising of MCF7(breast), NCI-H 460(lung), SF-268(CNS) [14].

One of its biological properties (antiviral) has gained immense attention due to the current outbreak of COVID-19 in December 2019 and has spread rapidly across the globe. A pandemic arises when a new type of virus or bacteria is detected in human beings and can transmit from one individual to another. These pathogens are highly contagious, and overtime, mutates, making it difficult to control and create an effective drug to combat these infections. In the year 2020, a new pandemic situation arose after the outbreak of the Ebola virus in the year 2013 (December) in Guinea [16,17], called SARS-COVID 19, which originated in Wuhan China [18]. It has spread globally in a short duration of time, affecting millions of people. It is very similar to bat coronavirus and is also believed to have originated from bats and transmitted to humans from wet animal markets in Wuhan, China [19]. Since the quest to prepare effective coronavirus drugs is still being developed and clinically tested, quinoxalines based heterocycles stand a fair chance to be examined on COVID-19 patients because of their impeccable property, an antiviral.

The biological applications of quinoxaline derivatives are broad and have aroused interest in the scientific community; therefore, this review paper will focus on synthesizing novel quinoxaline and its derivatives in the last two decades with straightforward synthetic routes. It also reviews their potentials to fight infections and has an open window to be tested on infectious and non-infectious diseases.

## 2. Synthetic Pathways to Prepare Quinoxalines via Cost-Effective and Green Synthetic Approach

### 2.1. Using Bentonite Clay K-10

The green organic synthesis approach is gaining much ground for research because it is environment friendly, and limitations like harsh reaction conditions, expensive reagents, and low yield can be eliminated. Clay is a very cheap, heterogeneous green reagent and is readily available. This concept was formulated by A. Hasaninejad et al. [20] to synthesize quinoxaline derivatives by using bentonite clay K-10 at room temperature. The reaction is progressed by mixing reactants (**1**) and (**2**) with bentonite clay and ethanol in a round bottom flask, as indicated in Scheme 1. After completing the reaction, it is carefully poured on a celite pad and washed with warm ethanol. The reaction mixture was concentrated to 5 mL, followed by dilution with 10 mL of water. It was allowed to stand undisturbed for 1 h. The clay was recovered after the formation of pure crystals and used again up to five times. This synthesis agrees with green chemistry protocols, and similar studies on the development of efficient and environmentally friendly methodologies in organic synthesis should be established. The results were briefed in Table 1 and presented the best results in 2.5 g and 3 g of clay, with a short reaction time of 20 min.

Different solvents were selected to decide the best solvent to synthesize 2,3-diphenylquinoxaline (**3**), as shown in Table 2. The best yield in a short time was noticed in ethanol.

Betonite clay K-10 overlays the way for green chemistry organic synthesis, with exceptionally mild conditions such as short reaction time, high yield, low cost, and simple experimental and isolation procedures.

### 2.2. Using Phosphate Based Heterogeneous Catalyst (MAP, DAP, TSP)

Since the development of green chemistry, enormous research has been conducted to produce clean and environmentally friendly chemical processes [21] to protect human health and its surroundings. Heterogeneous catalysis has been used extensively in green organic chemistry due to its recyclable properties and low energy consumptions. This property was utilized by Malek B et al. [22] to prepare quinoxaline derivatives by utilizing phosphate-based heterogeneous catalyst fertilizers such as mono-ammonium phosphate (MAP), di-ammonium phosphate (DAP), or triple-super phosphate (TSP). In order to test the reliability of the protocol shown in reaction Scheme 2, various aryl-1,2-diamine (1 mmol) (**4**) was condensed with benzyl (1 mmol) (**5**) in EtOH (2 mL) and phosphate-based catalyst MAP, DAP, or TSP (0.0006 g). The resultant product (**6**) was recrystallized using hot ethanol. The catalyst is retrieved from the reaction mixture by simple filtration, washed with hot ethanol, and dried for 6 h at 80 °C.

### 2.3. Using Lanthanide Reagent (CAN)

One of the lanthanide reagents, Cerium(IV)ammonium nitrate (CAN), has gained much attention in organic chemistry due to its low cost, miscibility in water, and high reactivity. Using CAN as a catalyst in organic synthesis is considered a safe green chemistry approach [23]; therefore, Yao et al. [24] prepared biologically important quinoxalines using a catalytic amount of CAN in water. The reaction between substituted benzil (**7**) and derivatives of ortho-phenylenediamine (**8**) using a catalytic amount of CAN either in methyl cyanide or any protic solvents produced quinoxaline derivatives (**9**), as presented in Scheme 3. The product was achieved in 20 min without any side products. Quinoxaline’s derivatives (**9**) can be evaluated for their antimicrobial properties like antiviral, antibacterial, antifungal, and many more. Synthesis of quinoxalines using lanthanides as a catalyst should be explored as a green chemistry approach.

### 2.4. Using Fe as a Catalyst

One of the essential examples of the N-heterocyclic compound is pyrrolo[1,2-α]quinoxalines, which are often found in nature, with suitable pharmacological activities. Due to its incredible applications, pyrrolo[1,2-α]quinoxalines has gained immense attention for synthesis via different routes. The synthetic pathways for the preparation of pyrrolo[1,2-α]quinoxalines by using Fe catalysis with 1-(2-aminophenyl)pyrroles(10) and cyclic ethers has been developed by Zheyu et al. [25]. Synthesis of pyrrolo quinoxalines was cultivated because of the low-cost and ease of availability of reagents. Reaction Scheme 4 establishes the synthetic route for both cyclic and linear ethers. Initially, the reaction was focussed on 1-(2-aminophenyl)pyrrole (**10**) with THF, Fe as a catalyst, and tert-butyl hydroperoxide(TBHP), stirring at rt for ten hours. The desired product [11,12] was obtained with a 46% yield. The percentage was increased to 94% by the addition of CF_3_SO_3_H as an additive. To conclude, this reaction proceeds by the cleavage of the C–O bond of cyclic ethers to get C–C and C-N bonds to synthesize pyrrolo[1,2-α]quinoxalines.

### 2.5. Using Fluorinated Alcohols (HFIP)

The fluorinated alcohols have gained massive consideration in organic reactions due to their low nucleophilicity, high polarity, strong hydrogen bond donating ability, and the ability to solvate water [25]. Fluorinated alcohols can also stabilize the helix conformations of proteins [26]. The property of fluorinated alcohol to prepare quinoxaline derivatives has been developed by Khaskar et al. [27]. The reaction was progressed in hexafluoroisopropanol(HFIP), with benzil (**13**) and orthophenylenediamine (**14**) at room temp for 1hour to yield 2,3-diphenylquinoxaline (**15**) with a 95% yield, as shown in Scheme 5. The condensation of aryl 1,2-diamines with 1,2-dicarbonyl compounds in ethanol or acetic acid is a common approach [28,29], which has problems related to long reaction time, high reaction temperature, low yield, use of toxic organic solvents, and many more. It also has an exemplary green chemistry aspect since the HFIP can quickly recover and be recycled at least five times without any significant activity change. This procedure can be applied in the large scale preparation of quinoxalines.

### 2.6. Using Pyridine as a Catalyst

Quinoxaline derivatives can be synthesized using 1,2-phenylenediamine and two carbon synthones such as α-dicarbonyls, α-halogeno carbonyls, α-hydroxy carbonyls, α-azo carbonyls, epoxides, and α, β-dihalides. Quinoxaline derivatives can also be prepared via the reaction of phenacyl halides with phenylene-1,2-diamine, which is a condensation-oxidation reaction in the presence of a catalyst and/or medium, which formed the basis of the synthesis of quinoxalines by Narsaiah et al. [30] using pyridine as a catalyst. As demonstrated in reaction Scheme 6, an equimolar quantity of 1,2-diaminobenzene derivatives (**16**) and phenacyl bromide (**17**) is reacted in THF in the presence of pyridine at room temperature. The reaction was over in 2 h to get the desired product 2-phenyl quinoxaline (**18**) in the right quantity. The 10% mole of pyridine exhibited optimum results to prepare 2-phenyl quinoxaline derivatives. Further evaluation of the pharmacological conduct of quinoxalines (**18**) should be assessed. Since the reaction (Scheme 6) can be completed at room temperature and is a one-step procedure, more preference can be given to study biological properties like antifungal, antibacterial, antiviral, anticancer.

### 2.7. Using a Solid Acid Catalyst, TiO_2_-Pr-SO_3_H

The rate of organic reactions can be accomplished by using catalysts. Recyclable catalysts have gained immense focus by organic researchers, such as solid acid catalysts like TiO_2_-Pr-SO_3_H, due to green chemistry synthesis. The properties of TiO_2_-Pr-SO_3_H were used by Atghia and Beigbaghlou [31] to prepare quinoxalines because the reaction can progress at room temperature with reduced reaction times. Many catalysts have been used in past years to prepare quinoxalines like alumina [32], montmorillonite K-10 [33], sulphated TiO_2_ [34], clayzic [35], Zirconium(IV) modified silica gel [36], PEG-400 [37], heteropolyacid [38], ZrO_2_/MxOy/MCM-41 [39], cellulose sulfuric acid [40] and Ga(OTf)_3_ [41]. Even though significant, these procedures have limitations like long reaction times and potential hazards in catalyst preparations. The reaction Scheme 7 elucidates a simple one-step synthesis of quinoxalines. The reaction between substituted 1,2-phenylenediamine (1 mmol) and benzyl (1 mmol) is evaluated under varied solvents like EtOH, THF, MeCN, EtOAc, and toluene, as well as in solvent-free conditions. The top results were attained in TiO_2_-Pr-SO_3_H (10 mg) and EtOH, with a 95% yield within 10 min.

### 2.8. Using the Catalytic Amount of Acetic Acid and Aldehydes

Pyrrolo[1,2-a] quinoxalines are a vital class of heterocyclic compounds and are contemplated as biological heterocycles. It has gained prominence because substitution at C-4 pyrroloquinoxalines boosts its physiological properties like anticancer, antiviral, and antiproliferative effects. Various synthetic pathways have been developed to prepare pyrroloquinoxalines [42,43,44] because it is crucial in drug discovery. For this reason, an efficient method was established by Allan et al. [45], which is an acid-catalyzed reaction for the synthesis of 4-aryl substituted pyrrolo[1,2-a] quinoxalines (**24**). This one-pot reaction involves imine formation, followed by cyclization and air oxidation. As disclosed in reaction Scheme 8, the reaction progresses by cyclization of 1-(2-aminophenyl)pyrroles (**22**) with a substituted aldehyde (**23**) in acetic acid and methanol as solvent. The reaction is refluxed for 8 h at 60 °C. The established reaction Scheme 8 is a classic example of the Pictet-Spengler reaction using catalytic amounts of acetic acid.

### 2.9. Using an Excess of Secondary Amines in Boiling Benzene

Interferons (IFNs) are glycoproteins made or released by host cells to treat pathogens such as bacteria, pathogens, tumor cells, and especially viruses. Virally infected cells produce and release proteins called interferons, which prohibit the multiplication of infected cells. Although available, interferon drugs have side effects like low potency in long-term usage [46,47,48]. In the quest to prepare new antiviral drugs, Shibinskaya et al. [49] synthesized a series of indoloquinoxalines (**28**) having low toxicity as interferon inducers and anti-VSV activity (Vesicular stomatitis virus). The target compound (**28**) was synthesized by following Scheme 9. 1-(2-Bromoethyl)-indole-2,3-dione (**26**) was made via isatin alkylation of (**25**) with an excess of dibromoethane in DMF at room temperature in the presence of K_2_CO_3_. Further condensation of (**26**) with 1,2-diaminobenzene in boiling acetic acid forms 6-bromoethyl-6H-indolo-[2,3-b]quinoxaline (**27**). The final compound (**28**) was achieved by aminodebromination of (**27**) by an excess of secondary amines in boiling benzene.

All the synthesized indoloquinoxalines exhibited interferons inducing activities.

### 2.10. Using 6-chloro-7-fluoro-1,2-Diaminobenzene with Diketones

AIDS, which is Acquired immune deficiency syndrome, is caused by an immune deficiency virus called HIV. HIV weakens the immune system and makes the body susceptible to various infections [50]. Even though various anti-HIV drugs are available, they become resistant over time, which is the foremost hurdle in the treatment process. To overcome the virus-resistance, the utilization of integrase inhibitors with the anti-HIV drug has shown certifying results. To date, three drugs Raltegravir, Elvitegravir, and Dolutegravir, are available as intergrase inhibitors. Resistant to Raltegravir and Elvitegravir has been reported with no side effects [51]; therefore, the development of an anti-HIV drug is essential. In the pursuit to prepare the anti-HIV agents, Patel et al. [52] applied two ligands-based drug approaches: pharmacophore forming and 3D QSAR. The results were merged to prepare quinoxalines and their derivatives and were scrutinized for antiviral properties.

A new sequence of 6-chloro-7-fluoro quinoxaline derivatives was formed, as indicated in Scheme 10. The first step entails the protection of the amino group of 3-chloro-4-fluoro benzamine (**29**) via acetylation reaction to give an intermediate n-(3-chloro-4-fluorophenyl)acetamide (**30**) with 89% yield. Further, the nitration reaction is carried out to yield another intermediate *N*-(2-nitro-4-fluoro-5-chlorophenyl)acetamide (**31**) with a 90% yield. The reduction of the nitro group to an amino group ensued in the formation of 1,2-diamino-4-chloro-4-fluorobenzene (**33**). The product was recrystallized with dichloromethane with a 90% yield. The diamino group is engaged in ring closure with various diketones (**34**) to generate 6-chloro-7-fluoro quinoxalines (**35**), and final purification is achieved via column chromatography. The synthesized compounds (**35**) were assessed for their anti-HIV activity and bulky substitutions at C-2, C-3 exhibited the best anti-HIV agents compared to less bulky substitutions. Furthermore, the fluoro group at C-2, C-3 also recorded as good anti-HIV agents.

## 3. Synthetic Pathways to Prepare Biologically Active Quinoxaline Derivatives

### 3.1. Synthesis of Quinoxalin-2-Mercaptoacetyl Urea as Antiviral Agents

A pandemic situation in western Africa infected almost 28,000 individuals in 2014–2015 (WHO: Ebola situation report 2015). Even though vaccines are existing, it adversely affects some patients; therefore, it becomes requisite to formulate new antiviral drugs and vaccines. A series of quinoxaline-2-mercaptoacetyl urea analogs was devised by Loughran et al. [53] and tested for their antiviral properties on Marburg, Ebola VP40 VLP budding assays in HEK293T cells.

The reaction Scheme 11 depicts the formulation of the target compound (**38**) by alkylation of quinoxaline thiols (**36**) with α-chloroacetamidoureas (**37**). The alkylating agents (**37**) were attained via the reaction of commercially available anilines or heteroaromatic amines (R-NH-Ar) with commercially available chloroacetyl isocyanate.

The synthesized product quinoxalin-2-mercaptoacetyl urea showed improved potency as an RNA viral egress inhibitors and inhibited live virus egress (VSVM40).

### 3.2. Synthesis of Quinoxaline Nucleosides as Anti-HIV Agents

Quinoxalines have various pharmacological applications such as anti-inflammatory, antidepressant-tranquillizing, antitumor, and anti-hepatitis B virus (HBV) activity. The biological significance of quinoxaline derivatives prompted Ali et al. [54] to synthesize some homo unsaturated acylnucleosides quinoxaline derivatives.

Numerous acylonuclosides analogs have chemotherapeutic antiviral activities, and structure-activity relationships in acylnucleosides play a crucial role in their antiviral target enzymes(phosphorylation) [55].

Scheme 12 illustrates the layout of a new series of acyclic quinoxaline nucleosides. The quinoxaline base (**39**) with (*R*)-2,2-dimethyl-1,3-dioxolan-4-ylmethyl-p-toluenesulfonate (**40**) in the presence of NaH/DMF gives (**41**) which on further acid hydrolysis yields 1-(2,3-dihydroxy propyl)-6,7-dimethyl-quinoxaline-2-one (**42**). The target product (**42**) showed inhibition of HIV-1 with an EC50 value of 0.15 ± 0.1 µg/mL and a therapeutic index of (SI) 73.

### 3.3. Synthesis of Penta-1,4-dien-3-one Oxime Containing a Quinoxaline Nucleus as Antiviral Agents

A well-known plant virus is known as the Tobacco mosaic virus (TMV), which is known to infect nine plant species, including tobacco, tomato, pepper, and cucumbers [56]. Controlling plant diseases, once infected, is a severe problem faced by agriculture industries. Traditional antimicrobial agents and plant virucides have caused resistance in plant pathogens; therefore, a synthesis of safe agricultural chemicals is always in demand. The agricultural chemicals derived from natural products are environmentally friendly and have unique bioactivities. Xia et al. [57] synthesized a series of penta-1,4-dien-3-one oxime comprising a quinoxaline moiety, equivalent to curcumin isolated from plant Curcuma Longa L. The synthesized compounds displayed antibacterial and antiviral activities, with distinct activity against TMV. As revealed in Scheme 13, 2-chloroquinoxaline (**43**) and penta-1,4-dien-3-one oxime ether (**44**) were stirred for 30 min and refluxed for 4 h at 80 °C to get the target compound (**45**). The product is recrystallized with acetonitrile and dichloromethane. Some compounds demonstrated significant, beneficial, protective, and inactivation activity against TMV, with a 50% effective concentration (EC50) of 287.1, 157.6, and 133.0 mg mL^−1^, respectively. The findings were superior to or equal to those of ningnanmycin (356.3, 233.7, and 121.6 mg mL^−1^, respectively).

### 3.4. Synthesis of Methyl-2-[3-(3-phenylquinoxalin-2-ylsulfanyl)propanamidoalkanoates and N-Alkyl-3-((3-phenyl quinoxalin-2ylsulfanyl)propanamides as Antitumor Agents

Quinoxalines have immense anticancer properties and experimented with in many research projects. Compounds with quinoxaline nucleus have found ground in many anticancer agents. Quinoxaline derivatives validate the right anticancer action through separate mechanisms, involving tyrosine kinase inhibition, C-MET kinase inhibition, induction of apoptosis, tubulin polymerization inhibition, and selective induction of tumors hypoxia [58].

Rayes et al. [58] synthesized new sets of quinoxaline moieties coupled with amino acids, or *N*-alkylamines are shown in Scheme 14 and Scheme 15 to evaluate their antitumor activities. Synthesis of (**47**) is accomplished by reacting phenylquinoxalines-2(1*H*)-thione (**46**) and triethylamine, with acrylic acid derivatives under reflux for 4–6 h. The attained compound (**47**) was treated with hydrazine hydrate in ethyl alcohol afforded (**48**) with 88% yield, as depicted in Scheme 15. The reaction was further progressed with NaNO_2_ and HCl in an ice bath for 15 min. The azide derivative (**49**) is extracted with ethyl acetate. Furthermore, (**49**) reaction with amino acid methyl ester hydrochlorides in the presence of triethylamine yielded (**50**). Likewise, the azide derivative was also reacted with alkyl amines to procure (**51**).

The synthesized compounds (**50**,**51**) manifested unique anticancer properties with IC50′S in the low molar range. The most active compound exhibited IC50′S of 1.9 and 2.3 µg/mL on the HCT-116 and the MCF-7 cell lines, respectively, compared to reference drug doxorubicin (IC50 3.23 µg/mL). Therefore, it is indispensable to expand this area of study, to prepare quinoxaline derivatives via the thiation of novel 3-phenylquinoxalin-2(1*H*)-one to get potent anticancer drugs.

### 3.5. Synthesis of Quinoxaline Derivatives as Anticancer Agents

Cancer is still a cause of significant deaths globally and is contemplated as a significant health problem. Quinoxalines are an essential base for anticancer drugs and have proved to possess selective adenosine triphosphate (ATP) inhibitors in many kinases [59]. The antitumor activity of kinase inhibitors comprising diaryl urea has earned immense attention because of its binding mode. Owing to its beneficial anticancer properties, Abouzid et al. [60] synthesized an array of quinoxaline-based compounds with amide, urea, thiourea, and sulfonamide moieties. Quinoxalines is derived from the reaction of o-phenylenediamine and α-keto-carboxylic acids [61]. The chloroquinoxalines (**52**) were further refluxed with m-aminobenzoic acid in n-butanol, the solution was left to cool, and the surplus of m-aminobenzoic acid was removed, dissolving in 5% NaOH and precipitated by dropwise addition of concentrated HCl. The intermediate (**53**) was activated as acid chloride derivatives (**54**) by refluxing with an excess of thionyl chloride [62] and then followed by treatment with the appropriate amount of aromatic amines (aniline, p-chloroaniline, p-methoxyaniline) in dichloromethane, yielding the final product (**55**) [63], as indicated in Scheme 16.

In another route, (Scheme 17) 2,3-dichloroquinoxaline was refluxed with p-phenylenediamine to get an amide intermediate (**56**). The amide derivative on further treatment with phenyl isocyanate, phenyl isothiocyanate in dry toluene resulted in thiourea and urea quinoxaline derivatives (**57**,**58**) [64].

The prepared quinoxaline scaffolds were screened for their cytotoxicity against three tumor cell lines. Some of the prepared quinoxaline derivatives (**58**) had activity against human colon carcinoma (HCT 116) cell lines (IC50 = 2.5 µM). Therefore, it is an exemplar compound for further studies for the optimization and development of anticancer drugs.

### 3.6. Synthesis of 4-(2-Methyl quinoxaline-3-yloxy)benzaldehyde and N-((4-(2-methylquinoxaline-3-yloxy)phenyl)methylene)-4-Substituted Benzenamine as Antibacterial/Antifungal Agents

An aim to synthesize new Schiff bases containing quinoxaline moieties using 2-chloro-3-methylquinoxaline as a reactant was created by Singh et al. [65]. The C2 chlorine is replaced by an ether linkage affixed to a benzene ring having a free aldehyde or amine group. The reaction Scheme 18, expounds the synthetic pathway to prepare 4-(2-methyl quinoxaline-3-yloxy)benzaldehyde (**60**) and *N*-((4-(2-methylquinoxaline-3-yloxy)phenyl)methylene)-4-substituted benzenamine (**63**) using 2-chloro-3-methyl quinoxaline(**59**). A mixture of (**59**) and 4-hydroxy benzaldehyde was refluxed in acetonitrile for 30 h to afford (**60**) as an intermediate. The intermediate (**60**) on treating with assorted substituted amines yielded (**62**). In another pathway, the nucleus (**59**) is treated with 4-amino phenol and refluxed for 30 h in acetonitrile to get a second intermediate (**62**). The final quinoxaline derivative (**63**) is generated by the reaction of (**62**) with substituted aromatic aldehydes. All compounds were purified using ethanol, with a 60–70% yields.

All synthesized compounds were checked for their antimicrobial activities. Most of the newly formed Schiff bases [17,19] were screened for antibacterial and antifungal properties with promising results. Therefore, it keeps an open window to synthesize more novel quinoxalines and study for their anti-infectious properties.

### 3.7. Synthesis of 2-(5-Arylthiazolo[2,3-c][1,2,4]triazol-3-yl)quinoxaline Derivatives as TP Enzyme Inhibitor

Over the last years, several derivatives of the six-membered ring with two nitrogen atoms have been synthesized and were reported to show inhibition for Thymidine phosphorylase(TP) [66], which is one of the classes of enzymes involved in catabolism for both prokaryotic and eukaryotic organisms [67,68,69]. The low-cost synthesis and large-scale preparations of 2-(5-arylthiazolo [2,3-c] [1,2,4] triazol-3-yl)quinoxalines were synthesized by Almandil et al. [70]. The quinoxaline-2-carbohydrazide (**64**) is treated with potassium thiocyanate(KSCN) in the presence of an acid, followed by basic solution treatment to develop (**65**), as set in reaction Scheme 19. Furthermore, the formed intermediate (**65**) was reacted with several substituted phenacyl bromide to afford the crude compound (**66**). The product was washed with water and purified using hot methanol with a 75–80% yield. The synthesized quinoxaline derivatives (**66**) were screened for inhibitory potential against TP enzymes. They displayed a range of inhibition with IC50 between 3.50 ± 0.20 to 56.40 ± 1.20 µM as compared to standard 7-Deazaxanthine with IC50 = 38.68 ± 1.12 µM.

### 3.8. Synthesis of Spiro[thiadozoline-quinoxaline] Derivatives as Antibacterial Agents

1,3-dipolar cycloaddition is a subject of intense research owing to their great synthetic value. It is a synthesis of five-membered heterocyclic compounds with significant physiological properties. Synthesis of spiro[thiadiazoline-quinoxaline] derivatives was developed by Mokhtar et al. [71] by 1,3-dipolar cycloaddition of 3-methylquinoxaline-2-thione and their *N*-alkylated derivatives. As presented in reaction Scheme 20, thionation of N-alkyl quinoxaline (**67**) with phosphorous pentasulphide(P_4_S_10_) in refluxing pyridine formed varieties of alkyl quinoxaline derivatives (**68**). On further investigation, 1,3-dipolar cycloaddition was performed on 1-ethyl-3-methyl quinoxaline-2-thione (**69**) with an equimolar quantity of diphenyl hydrazonoyl (**70**). The reaction mixture is refluxed in dry tetrahydrofuran(THF) in the presence of triethylamine (Et_3_N). One cycloadduct (**71**) was obtained on a dipolarophillic group C=S. Diphenyl nitrile imine ylide(DPNI) is generated in situ from diphenyl hydrazonoyl chloride.

The newly derived spiro[thiadiazoline-quinoxaline] exhibited antibacterial activities. As reported, the ethyl group’s presence enhanced the properties by 64 µg/mL against streptococcus fascines to 128 µg/mL against S.aureus. Further studies on spiro[thiadiazoline-quinoxaline] should be examined on other pathogens.

### 3.9. Synthesis of Bistetrazoloquinoxalines as Antiallergic Agents

Tetrazoles are a class of heterocyclic compounds comprising the five-membered ring with four nitrogen atoms and one carbon atom. The fusion of tetrazoles with quinoxalines displays remarkable biological activities; for example, 4-chlorotetrazolo-(1,5-a) quinoxalines obstruct mast cell-mediated allergic reactions [72]. The synthesis of bistetrazolo-[1,5-a; 5′, 1′-c]-quinoxalines was designed by Prasanna. et al. [73] using one-pot three-component synthesis. Scheme 21, displays the reaction of 2,3-diketoquinoxalines (**72**) with POCl_3_(Phosphorous oxychloride) and sodium azide, resulting in the formation of bistetrazoloquinoxalines (**73**). The reaction mixture was refluxed for 2–3 h, cooled, poured in crushed ice, and recrystallized by rectified spirit. The starting material (**73**) is synthesized by condensation of o-phenylenediamine with oxalic acid in 4N HCl, applying Philip’s procedure [74]. Therefore, good yields, short reaction times, and ease of preparations certify an ideal reaction to prepare bistertazoloquinoxalines.

### 3.10. Synthesis of 4-{4-[2-(4-(2-Substituted quinoxaline-3-yl)piperazin-1-yl)ethyl] phenyl} Thiazoles Antipsychotic Agents

Schizophrenia is a lifelong psychotic disorder that affects a small percentage of people and is considered a significant problem to existing health diseases worldwide [75]. The development of antipsychotic drugs was first introduced in the 1950s to manage schizophrenia, which was the first breakthrough in this field [76]. In a novel quest to prepare antipsychotic drugs with minimum side effects, Chandra Sekhar synthesized a series of 4-{4-[2-(4-(2-substituted quinoxaline-3-yl)piperazin-1-yl)ethyl] phenyl thiazoles (81), a novel atypical antipsychotics [77]. The reaction Scheme 22 exhibits the formation of 2-Chloro-3-(piperazin-2-yl)quinoxaline (**76**) and 2-methoxy-3-(piperazin-2-yl)quinoxaline (**78**), which is one of the prime components to prepare antipsychotic drug.

The chloro compound (**74**) on reacting with piperazine (**75**) in the presence of anhydrous Na_2_CO_3_ yielded (**76**). Furthermore, 2-chloro-3-methoxy quinoxaline (**77**) was attained on stirring with methanol in the presence of phase transfer catalyst triethyl-benzyl ammonium chloride(TEBAC) at room temperature, which on further reaction with piperazine in acetonitrile yielded 2-methoxy-3-(piperazin-2-yl) quinoxaline (**78**).

In the last step, the prepared piperazinyl quinoxalines (**79**) and chloroethyl phenyl thiazoles (80) in equimolar quantity in the presence of Na_2_CO_3_ and a catalytic amount of KI in DMF, afforded4-{4-[2-(4-(2-subsitutedquinoxalin-3-yl)piperazin-1-yl)ethyl] phenyl} thiazoles (**81**) as the final product, as shown in Scheme 23. The synthesized compounds (**81**) were evaluated for their antipsychotic activities in animals as models. Some of the synthesized quinoxalines (**81**) were more active than standard drug Risperidone, hence satisfying all criteria to be classified as an antipsychotic drug according to Meltzer’s classification [78]. Further studies of this synthesized drug (**81**) are crucial for a breakthrough in schizophrenia and other mental disorders in patients.

### 3.11. Synthesis of Quinoxaline-2,3(1H,4H)-Dithione Derivatives as Antimicrobial Agents

The synthesis of quinoxaline-2,3-(1H,4H)-dithione was summarized by Baashen 2018. [79], which is a key method to prepare quinoxaline dithione derivatives in heterocyclic chemistry. Quinoxaline-2,3(1*H*,4*H*)-dithione (**59**) can be synthesized by using a wide range of thionating agents such as phosphorous pentasulfide, thiourea, and sodium hydrogen sulfide [80,81] as illustrated in Scheme 24. The thionation of quinoxaline-2,3(1H,4H)-dione (**82**) with a crystalline zwitterionic dipyridine-diphosphorus pentasulfide complex in refluxing pyridine for 1 h produces quinoxaline-2,3(1*H*,4*H*)-dithione (**84**) with 83% yield. The reaction of 2,3-dichloroquinoxaline (**83**) with thiourea affords (84) in moderate to good yields. The best yield (86%) of quinoxaline-2,3(1*H*,4*H*)-dithione was synthesized by treatment of (**83**) with sodium hydrogen sulfide(NaSH) in ethanol under reflux for 5 h followed by neutralization with acid.

The synthesized quinoxaline dithiones were further modified to provide different quinoxaline based heterocyclic compounds. The synthesized compounds should be investigated for their antimicrobial properties like antiviral, antibacterial, antifungal. The depicted Scheme 25 exemplifies the synthesis of quinoxaline dithione derivatives.

### 3.12. Synthesis of [1,2,4]triazolo[4,3-a]quinoxaline and bis([1,2,4]triazolo)[4,3-a:3′,4′-c]quinoxaline as Anti-Tumor Agents

A new series of [1,2,4]triazolo[4,3-a]quinoxaline and bis([1,2,4]triazolo)[4,3-a:3′,4′-c]quinoxaline derivatives have been synthesized by Ibrahim et al. [82]. Furthermore, they were biologically evaluated for their cytotoxic activities against three tumor cell lines (HePG-2, Hep-2, and Caco-2). Further studies were planned to evaluate their topoisomerase 11(Topo 11) inhibitions and DNA intercalating affinities as prospective antiproliferative activities. As shown in reaction Scheme 26, the target compounds **91**, **92**, **93** have been prepared by stirring 2, 3-dichloroquinoxaline with hydrazine hydrate to afford **89,** which on subsequent heating with triethylorthoformate gives 4-chloro [1,2,4]triazolo[4,3-a]quinoxaline (90). The intermediate 90 afforded three different quinoxaline derivatives on treatment with hydrazine hydrate to get 4-hydrazinyl-[1,2,4] triazolo[4,3-a]quinoxaline (**91**), with alkyl amines to get (**92**) and aliphatic alcohols to produce (**93**).

In another synthetic route, as demonstrated in Scheme 27, the synthesized compound (**91**) on treatment with an alcoholic solution of CS_2_ and KOH produced bis [1,2,4] triazolo[4, 3-a: 3′, 4′-c]quinoxaline-3-thiol (**95**). The Potassium salt (**96**) on treatment with ethyl-4-bromobutyrate in dry DMF afforded (**97**).

Compounds 91 and 95 demonstrated the highest activities against the observed cell lines (tumor cells) with IC50 values ranging from 0.29 to 0.90 µM comparable to doxorubicin (IC50 ranging from 0.51 to 0.73 µM).

### 3.13. Synthesis of Thiadiazino and Thiazolo Quinoxaline Derivatives as Antibacterial/Antifungal Agents

Quinoxalines are pharmacologically useful agents like Dazoquinast(antiallergic), U8044(antidepressant, anxiolytic), LU 73,068(anticonvulsant glycine/NMDA), and extant in antibiotics, such as Echinomycin, which is known to inhibit the growth of gram-positive bacteria [83]. Fused quinoxalines have been reported to have antimicrobial activities; therefore, Ammar. et al. [84] synthesized new quinoxaline derivatives incorporating aromatic thiadiazine or thiazole moieties fused to 6-(morpholine-4-sulfonyl)-1,4-dihydroquinoxaline to acquire improved antimicrobial agents. As depicted in the reaction Scheme 28, 6-Morpholinosulfonyl-2,3-dichloroquinoxaline (**98**) was synthesized in good yields by chlorinating 6-morpholinosulfonylquinoxalindione (**97**) utilizing phosphorus oxychloride. The intermediate (**98**) were further subjected to different synthetic routes to devise potent quinoxaline derivatives.

On treatment with 1,4-binucleophiles (thiocarbohydrazide (i) and pyazole-1-carbothiohydrazide (ii)), and 2,3-dichloroquinoxaline derivative (**98**) produced a single product which was formulated as 3-hydrazinyl-7-(morpholinosulfonyl)-1H-[1,3,4]thiadiazino[5, 6-b]quinoxaline(99) and 3-methyl-1-(7-(morpholinosulfonyl)-1H-[1,3,4]thiadiazino[5, 6-b]quinoxalin-3-yl)-1H-pyrazol-3-ol(100).

In a similar manner, 2, 3-dichloro derivatives (98) was treated with 4-amino-5-methyl-4H-1,2,4-triazole-3-thiol (iii) in refluxing ethanol provided a single product, 2,3-methyl-9-morpholino- sulfonyl-5H-[1,2,4]triazolo [2, 3, 3, 4] [1, 3, 4]thiadiazino[5, 6-b]quinoxaline (**101**).

The synthesized compounds **98**, **99**, **100**, **101** exhibited bacterial and fungal activity with results comparable to that of Norfloxacin. The synthesized compounds had significant MICs value (1.95–31.25) μg/mL, comparable to that of Norfloxacin (1.25, 0.78, 1.57, 3.13 μg/mL).

### 3.14. Synthesis of 6-[(het)arylthiomethyl]quinoxaline Derivatives as Antiviral Agents

Enteroviruses (EVs) are viral pathogens belonging to a class of Picornaviridae family. Coxsackievirus A and B, poliovirus, echoviruses are Enteroviruses that causes infection in man after absorption and replication in a gastrointestinal tract [85]. The infection is typically asymptomatic but extends to secondary organs, leading to severe diseases [86,87]. The US food and drug administration have not assigned any specific antiviral agent for Enteroviruses; therefore, Sanna. et al. [88] prepared quinoxaline derivatives and evaluated them for their antiviral activity against representatives of ssRNA, dsRNA, and dsDNA viruses.

As shown in Scheme 29, 6-(bromomethyl)-2, 3-dimethoxyquinoxaline (102) and benzenethiol derivatives or pyridine-2-thiol (103) were stirred in dry DMF, in Cs_2_CO_3_ at 70 °C for 2.5 h. After cooling and dilution with water, light-colored powders were attained and purified by EtOH/H_2_O. The derivatives of 6-[(het)arylthiomethyl]quinoxalines (**104**) were formed, showing potent antiviral activity against coxsackievirus B5, with EC50 in the sub-micromolar range (0.3–0.06 µM).

### 3.15. Synthesis of Quinoxaline-2-carboxylate 1,4-dioxide Derivatives as Antimycobacterium Tuberculosis Agents

Over the last two decades, many mono, di-N-oxides, 2-oxo derivatives of quinoxalines have been prepared and tested for their antimicrobial activities; for example, quinoxalin-2-ones have antifungal properties [89], and quinoxalin-1-oxides have antibacterial properties [90].

In the pursuit to obtain potent antimicrobial compounds, Jaso et al. [91] synthesized new 6(7)-substituted quinoxaline-2-carboxylate-1,4-dioxide derivatives and examined for their antituberculosis activity. Tuberculosis is an infection of Mycobacterium tuberculosis and is considered a leading cause of death in infectious diseases, especially in developing countries.

The development of new antitubercular compounds to improve the current chemotherapeutic antituberculosis treatments is necessary and beneficial.

As displayed in Scheme 30, the starting compound (**105**) (benzofuroxane, 5-substituted or 5,6-disubstituted benzofuroxane) was added to β-keto ester; the mixture was allowed to stand at 0 °C. Triethylamine was added dropwise and stirred in darkness for 1–3 days. The obtained crude solid or brown oil was precipitated and washed with diethyl ether and purified using ethanol. Quinoxaline-2-carboxylate 1,4-dioxide derivatives (**106**, **107**, **108**, **109**) were obtained and evaluated for in vitro antituberculosis activity, with EC90/MIC values ranging between 0.01 and 2.30.

### 3.16. Synthesis of [1,2,4]triazolo[4,3-a]quinoxaline Derivatives as Antiviral/Antimicrobial Agents

The triazoles have a great significance as antimicrobial agents and quinoxalines have known to possess immense biological activity against infections; therefore, a series of antiviral and antimicrobial agents containing triazoles fused with quinoxalines, [1,2,4] triazolo [4, 3-a]quinoxalines and their isosteres, pyrimido-quinoxaline, were synthesized by Henen. et al. [92].

The thioamide group is introduced to increase the bioactivity of [1,2,4] triazolo[4, 3-a]quinoxaline as an antiviral agent. Furthermore, [1,3,4] oxadiazole and [1,2,4]-triazole subunits enhances the antimicrobial activity, hence were incorporated in [1,2,4]-triazolo [4, 3-a]quinoxaline ring.

The synthetic pathway affording (**111**) was accomplished via reaction of 4-chloro-8-methyl-[1,2,4]-triazolo[4, 3-a] quinoxaline-1-amine(**110**) with mercapto oxadiazole or mercapto triazole in the presence of anhydrous K_2_CO_3_ (Scheme 31). The reaction mixture was stirred overnight and poured into ice-cold water and recrystallized using DMF/water. Similarly, in another route, the starting reactant (**110**), when treated with an appropriate amount of 3-(substituted benzylidene amino)-5-mercapto-1*H*-1,2,4-triazoles formed (**113**), and were purified using DMF/water. The antiviral compound (**112**) was synthesized by a mixture of 4-chloro-8-methyl [1, 2, 4] triazolo [4, 3-a] quinoxaline-1-amine (**110**) and appropriate isothiocyanate derivatives in EtOH, refluxed for 6 h.

The newly synthesized compound (**112**) was tested for its antiviral properties on African monkey kidney cells against the Herpes simplex virus. The tested compounds displayed a reduction in plaque(plaque-reduction assay was used) by 25% at 20 mg/mL.

## 4. Conclusions

Quinoxalines are an essential class of nitrogen-containing heterocycles with a wide range of physiological effects. Quinoxalines have gained much interest in medicinal chemistry, owing to their well-known biological activity to fight infectious and non-infectious diseases; therefore, simple synthetic routes using the green methodology, cost-effective methods, and some biologically significant pathways have been summarized in this review paper. The current epidemic situation in the world has prompted researchers to synthesize effective drugs to fight COVID-19. Ongoing trials are under-way to synthesize drugs and vaccines. Since quinoxalines have a wide range of applications to fight infectious diseases, it should be well studied and tested for their antiviral properties.

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
