# Peer review of "Novel Synthetic Routes to Prepare Biologically Active Quinoxalines and Their Derivatives: A Synthetic Review for the Last Two Decades"

_molecules, 2021, doi:10.3390/molecules26041055_

Round 1

Reviewer 1 Report

Molecules-1026107

Novel synthetic routes to prepare biologically active quinoxalines and its derivatives: a synthetic review for the last two decades

Hena and Abdulmalek in this review wanted to summarize several synthetic pathways for the obtainment of the quinoxaline scaffold and how this moiety is conserved in several drugs.

As a result, the review, after the introduction, is divided in two subchapters, the first is focused on the different synthetic routes while the second part would highlight how this moiety is conserved in different classes of molecules, showing peculiar therapeutic features.

The idea of the authors should be interesting and the manuscript after a strong and accurate revision could be satisfactory, even if several deep changes must be done.

First of all, the English needs a native speaker revision: there are too many mistakes, typos, error in sentence structure or singular words/verbs instead of plural, or vice versa. The title brings one of these errors (its instead of their) and this mistake is repeated several times along the abstract and the paper.

Here below just a few examples of English mistakes:

Page 2, Line 2 studied extensively instead of extensively studied

Page 2, Lines 3-4 Condensation reaction between 0rtho phenylenediamine and dicarbonyl compounds instead of condensation reaction between ortho phenylenediamine and dicarbonyl compounds

Page 2, Lines 7/10 A little modification in their structure gives rise to different moieties, which has a remarkable pharmacological effect of fighting different diseases with little side effects instead of A little modification in their structure brings to different moieties, which have a remarkable pharmacological effect in fighting different diseases with little side effects.

Furthermore, some words are used with the first capital letter without any understandable reason. I would suggest using only lower-case letters, despite proper nouns.

In addition to language changes, other strong changes must be done. The abstract should be rewritten from the beginning. The removal of the subsections is mandatory and the whole abstract should be more consistent, it must be efficient by itself.

Concerning the introduction, in my opinion the correlation between quinoxalines and covid-19 treatment is not so strong to be the main topic of this first section. Neither Remdesevir nor Favipiravir or Chloroquines have quinoxaline moiety in their structures, I would ask authors why this choice.

Moreover, concerning subchapters 2 and 3, some main common issues should be evaluated. All the Schemes and the relative captions should be written in the same way: for the molecules the authors should keep the same structure size and should avoid structures stretches. For the captions they should use the same font and the same size.

In Figure 2 diazine is too general and not correct to the reported structure: it is a 1,4-diazine, usually called pyrazine.

From Page 2 authors should write mL instead of ml.

In subchapter 2.1, in my opinion is quite unreasonable to speak in terms of g of Clay without mentioning the amounts of reagents. Authors should use mol or equivalents for any reagents/reactives, as done in some of the following subchapters.

Table 1 has a typo: AMOUNT OF CLAY IN, g. Moreover, are yield and time independent from the scale of the reaction?

In subchapter 2.2, 1,2-diamine and 1,2-dicarbonyl are not correct, since no variables are present: authors should use the correct names, please correct.

In subchapter 2.3, the 1,2-diamine is here called in a different way, please choose a way of mention them and keep it all along the paper. Authors should not call benzil, but differently substituted benzil, since there are substituents.

In subchapter 2.4, please correct formulas in which the numbers are not subscripts.

In subchapter 2.5, in the last sentence in page 5, via should be written in italic.

In subchapter 2.7, please check the caption of the scheme. Authors should level out all the variables calling them in the same way, may all R (R1, R2 and R3). Moreover, when there are several possible substituents, authors should specify which could be the possibilities.

In subchapter 2.8, please correct: Pyrrolo[1,2-a]quinoxalines are (not is) an important class of heterocyclic..

In subchapter 2.9, K2CO3: all the numbers should be in subscript (this mistake is present several times all along the paper)

Scheme 9: the structures are stretched

In subchapter 2.10, I do not understand the choice of the subtitle (Using diamino group with diketones), since mostly of the subchapters have these reagents, in my opinion a better subtitle should be used.

Finally, all section 3 should be rewritten in a different way. The authors wanted to highlight the further reactions that could be done on a quinoxaline scaffold, obtaining specific drugs or specific classes, useful in a determined treatment. Following this aim, I suggest dividing the subchapters calling them as the therapeutic class or the disease that should be addressed. Moreover, also in section 3 the captions and the schemes should be adapted (same structure size, no structures stretches, same font, same size).

Author Response

Dear reviewer,

Hope ur doing well.

I want to thank you for your precious time, in evaluating my paper.

All the critism has been checked and changed according to your recommendations.

Please find the attached file, of my responses.

Thanks,

Ms.Hena Khatoon

Reviewer 2 Report

The manuscript by Henal and Abdulmalek presents a literature review describing synthetic routes to biologically active organic compounds containing the quinoxaline moiety. The methods of quinoxaline ring formation based on heterocyclization reactions are reviewed. Also, modification and functionalization of starting compounds with the quinoxaline fragment are discussed. The literature cited is representative and up-to-date. The review will be interesting and useful to organic and medicinal chemists and pharmacologists.

Criticism:

1) The last sentence in the abstract contains a senseless phrase: "... so that heterocyclic compounds containing Quinoxaline nucleus is synthesized and investigated for its infectious diseases".

2) Introduction is too focused on coronaviruses and should be rewritten in a more balanced manner, as quinoxalines possess many other types of biological activity discussed further in the manuscript.

3) Tables in the manuscript do not contain headers.

4) It should be indicated in the text or in the header to Table 1, what kind of yield (preparative or analytical) is given in the last column.

5) In the paragraph before Scheme 2, amounts of a catalyst and a solvent along with reaction duration and temperature are given. However, amounts of the reactants are not indicated, although they would be of interest in this context.

6) The sentence before Scheme 4 contains a bad phrase: "addition of additives".

7) The penultimate sentence on Page 5 contains an incorrect phrase: "Fluorinated alcohol is a safe approach...".

8) The last sentence on Page 5 contains an incorrect phrase: "...preparations of Quinoxalines can be prepared...".

9) The sentence before Scheme 8 is senseless: "This reaction ... should be further evaluated for its biological properties".

10) The first line on Page 8: 'shibinskaya" should be replaced with "Shibinskaya" (starting from a capital letter).

11) Scheme 10: Compound (40) which reacts with (39) should be properly positioned within the scheme.

12) The last line on Page 11: The phrase "target reactant" is wrong in sense because the term "reactant" usually refers to a starting compound, not to a product.

13) The second paragraph on Page 12: The phrase "IC50'S in the low molecular range" is wrong in sense. "Molar range" should be used instead of "molecular range" in a more specific way: "nanomolar", "micromolar", etc.

14) In Section 3.16, the first line contains a senseless phrase: "The presence of triazole moiety has a great significance as antimicrobial agents...".

15) Not all the reaction schemes are referred to in the text of the manuscript.

16) A serious polishing of English language is needed throughout the manuscript.

I recommend acceptance of the manuscript for publication after major revision.

Author Response

Dear reviewer,

Hope ur doing well

I have made all the recommended changes, with English language editing.

Thanks for your time and precious comments.

Regards,

Ms.Hena Khatoon

Round 2

Reviewer 1 Report

The authors partially revised the paper: the abstract is now concise and quite good; the introduction has no misleading sentences and other punctual mistakes were corrected.

Nevertheless, the English needs a more accurate revision, since too errors are present, starting from the first sentence of the abstract, which has no sense: “Quinoxalines belongs to a class of N-heterocyclic compounds are important biological agents, and a significant amount of research activity has been directed towards this class” It should be corrected may be as “Quinoxalines, a class of N-heterocyclic compounds, are important biological agents, and a significant amount of research activity has been directed towards this class.”

Concerning schemes and captions, the captions of all the schemes should be written in the same font, same size and same way, while they are sometimes with all capital letters and sometimes not. Moreover, the molecules should be written with the same software and should have the same structure size, structures stretch should be avoided, and all the variables should have the same letter (R with different subscripts should be a good choice). In the pdf version of the actual manuscript schemes were replaced by some identical ones, is it an error?

Author Response

Dear Sir/Madam,

hope ur doing good. 

I have changed the first sentence of the abstract as recommended.

Extensive editing of the reaction pathways has been done by using the same font, size, capitalization has been avoided except in places necessary, for eg. In 2.2 MAP, DAP, TSP/ in 2.3 CAN / 2.5 HFIP and some more places, where short forms are in capital letters.

All the molecules are drawn via Chem Draw, use of other software is not done.

To avoid stucture stretches, I have used the same size without compression.

All variables are labelled as R with subscripts wherever needed.

I have added variables to some schemes as suggested by the reviewer

For eg, in scheme 8, 11, 19,21

I have not replaced any schemes, with identical ones. I have introduced variables

to make it more effective.

In case, there is any such replacement, can the reviewer specify, so that my mistake can be corrected.

I have done more editing of my english language to avoid any grammatical mistakes.

Finally, I thank the reviewer for his/her valuable comments.

Thanks,

Ms. Hena Khatoon

Reviewer 2 Report

The revised version of the manuscript was improved significantly. 

Some schemes are disproportionately squeezed in one dimension (e.g. 3, 5, 15, 26, 30, and others) and should be redrawn.

The manuscript can be accepted for publication after minor corrections.

Author Response

Dear sir/ Madam

Hope ur doing good.

I have corrected all the compression of schemes, by maintaining the original size.

I sincerely thank you for ur valuable comments, and to help me on this journey of my manuscript corrections

Thanks,

Ms.Hena Khatoon